# Anesthesia Resistant Memories in Drosophila, a Working Perspective

**DOI:** 10.3390/ijms23158527

**Published:** 2022-07-31

**Authors:** Anna Bourouliti, Efthimios M. C. Skoulakis

**Affiliations:** 1Institute for Fundamental Biomedical Research, Biomedical Sciences Research Center “Alexander Fleming”, 16674 Vari, Greece; bourouliti@fleming.gr; 2Department of Molecular Biology and Genetics, Democritus University of Thrace, 68100 Alexandroupolis, Greece

**Keywords:** memory, aversive memories, anesthesia resistant memory, massed conditioning, cold shock, Drosophila

## Abstract

Memories are lasting representations over time of associations between stimuli or events. In general, the relatively slow consolidation of memories requires protein synthesis with a known exception being the so-called Anesthesia Resistant Memory (ARM) in Drosophila. This protein synthesis-independent memory type survives amnestic shocks after a short, sensitive window post training, and can also emerge after repeated cycles of training in a negatively reinforced olfactory conditioning task, without rest between cycles (massed conditioning—MC). We discussed operational and molecular mechanisms that mediate ARM and differentiate it from protein synthesis-dependent long-term memory (LTM) in Drosophila. Based on the notion that ARM is unlikely to specifically characterize Drosophila, we examined protein synthesis and MC-elicited memories in other species and based on intraspecies shared molecular components and proposed potential relationships of ARM with established memory types in Drosophila and vertebrates.

## 1. Introduction

Learning and memory are adaptive vital functions that enable animals to predict outcomes based on prior experiences. These outcomes may be rewarding, such as the location of food, shelter, and mates, or beneficial by promoting avoidance of potentially harmful situations, places, and predators. The fruit fly *Drosophila melanogaster* is capable of multiple forms of aversive and reward learning and memory in both associative and non-associative scenarios. Through its powerful and facile molecular and genetic arsenal, Drosophila has been a cardinal contributor to our understanding of many molecular mechanisms that underlie these processes.

Of the various learning and memory assays, the more robust and well-understood are the associative, negatively, or appetitively reinforced olfactory paradigms. Two equally aversive odorants serve as the conditioned stimuli (CS), while pulses of electric footshocks, or sugarwater are the respective aversive and appetitive unconditioned stimuli (US). One of the two odorants, the CS+, is presented concurrently with the US, while the other (CS-) functions as an unpaired control (Figure 1A,C and Figure 2). Learning or memory of the CS+/US association is manifested as selective avoidance of the shock-associated odor as it would predict punishment [1]. In the appetitive assay, hungry flies are conditioned to associate an odor (CS+) to the presence of a sugar reward (US) and selectively approach the CS+ over a non-rewarded (CS-) odor [2].

Learning in these associative paradigms may be referred to as Immediate Memory (IM), or Short-Term-Memory (STM) because the time interval between training and testing of learning-dependent performance does not typically exceed 3 min. Perdurance of the learned information over time due to processes that maintain and consolidate it constitute memory of the event. The persistence of memory depends on training intensity as well as the nature of the information. For example, one-time association of a specific odor with a food reward leads to much longer-lasting memory than association of the same odor with multiple pairings to electric shock [3]. Memories are classed according to their persistence timeline that also reflects time-dependent differential engagement of molecular processes that mediate them. Middle Term Memory (MTM), also known as Intermediate Memory (ITM), can be detected 2–4 h after training [1,4]. Later on, about 6 h after training, Long Term Memory (LTM) emerges [1,4], which requires de novo transcription and translation and is the longest-lasting memory form since it can be maintained for weeks [5,6].

## 2. Anesthesia Resistant and Anesthesia Sensitive Memories

For US/CS associations to be maintained over time, processes generally referred to as memory consolidation are activated, probably soon after the learning episode [7]. Memory consolidation is a time-dependent process occurring over hours, days, or even longer in some species and requires protein synthesis [8,9]. Unconsolidated memories are labile and can be disrupted by various amnestic agents in all animals tested [8,9].

An effective tool in investigating the time course of memory consolidation in Drosophila was provided by the observation that acute exposure to 4 °C anesthetizes flies, followed by speedy recovery to apparently normally functioning animals minutes after removal from the cold. By definition, anesthetic agents lead to loss of consciousness. Although it is unclear whether the flies are truly anesthetized or simply immobile due to the cold, the treatment was called cold anesthesia. The presumptive linkage probably derives from the analgesic effects of cold, which has been widely used as a local treatment for injuries [10,11], and also the fact that both cold shock and anesthetics are considered amnestic agents [12,13,14,15,16]. Sadly, no bona fide anesthetics have been tested in olfactory conditioning paradigms in Drosophila as yet. Irrespective of whether it is actually anesthetic, the cold shock provides an advantageous experimental tool to determine whether memory is consolidated at any given time post-training. It is also advantageous that flies recover quickly, so the amnestic shock can be used even minutes before testing. This immobilizing cold shock immediately following a round of 12 US/CS footshock/odor pairings is totally amnestic, resulting in complete loss of the preferential CS+ avoidance (Figure 1B,D). It is also specific to newly formed associations and unconsolidated memories, as it does not have any effect if administered before conditioning.

The effect of the cold shock is partially alleviated if given 30–40 min post training in one version of the associative conditioning paradigm ([16] and Figure 1D). By 90–120 min post-training in the more commonly used negatively reinforced associative paradigm [17], although delivery of a cold shock reduces total memory, a significant portion remains. This memory, which is insensitive to the ostensibly anesthetic cold shock, is therefore named *Anesthesia-Resistant Memory* (ARM) [16], while the eliminated memory is referred to as *Anesthesia-Sensitive Memory* (ASM) [5,6,17]. Therefore, memory of the training event 2 h or so post-training of odor/shock associations consists of co-existing ARM and ASM (Figure 1D).

Significantly, because it persists the amnestic treatment, ARM must not be labile at that time and therefore it likely represents relatively quickly consolidating memory. In contrast, ASM represents memory of the association which is unconsolidated at that time of cold shock and therefore in a labile phase. Whether ARM and ASM represented the same memory in different stages of consolidation or distinct memories remained an open question until the discovery of one mutant named *radish*, presenting selective ARM elimination [18]. This argued that ARM and ASM represent distinct memories of the odor/shock association, apparently engaging different mechanisms and presenting distinct consolidation kinetics. In fact, recent evidence revealed that there is a Short-Term ARM (ST-ARM), which can be observed around 5 min after training the flies and likely forms in parallel with learning [19,20]. The ASM component ostensibly represents the slower consolidating, protein-synthesis-requiring LTM, which behaviorally [5,6] and physiologically [4] becomes apparent 6 h after conditioning.

## 3. Massed Conditioning Yields ARM and a Novel Protein Synthesis-Independent Memory

Stable protein synthesis-dependent LTM can be achieved by 5–10 cycles of 12 US/CS pairings per cycle of negatively reinforced olfactory conditioning delivered with a 15-minute rest between cycles (Spaced Conditioning-SC) as illustrated in Figure 2 [17]. This spaced protocol is not necessary in the case of reward conditioning, which pairs the odor to the presence of sucrose. In this appetitive conditioning assay, just one training cycle is adequate to induce LTM formation [21,22]. This makes it difficult to determine whether appetitive ARM can be formed. However, there is evidence indicating absence of this “reward-ARM” after massed training [23].

To elicit ARM formation, either of two protocols of aversive olfactory conditioning is typically used. One protocol emulates the original experiment that yields ARM [16] and involves cold shock exposure [17]. In this setting, flies are trained to associate an odor to electric shock by a single training cycle typically consisting of 12CS/US pairings. The flies are returned to their home vials in the dark and 2 h post-training are exposed to a 2–4 min cold shock on ice (Figure 1). They are then returned to their home vials in the dark and tested for memory of the US/CS association 1 hour later. An alternative protocol, which does not involve cold shock, requires usually 5 or up to 10 consecutive training cycles as for those that yield LTM, but delivered without the rest interval between them (Figure 2, massed conditioning—MC). The massed protocol is preferable if memory is to be tested at 24 h or later after training as the intensive training leads to longer-lasting effects.

MC-yielded memory was thought to be anesthesia resistant based on two main points. First, like ARM, MC-induced memory is protein synthesis-independent as it is not susceptible to the protein synthesis inhibitor Cyclohexamide (CXM) [24], which disrupts SC-elicited LTM [17]. In addition, the mutation *radish*, which disrupts cold-shock insensitive ARM also disrupts MC-induced memory [17]. Although MC-yielded memory is often referred to as ARM, or the terms are used interchangeably, it has been unclear until recently whether MC elicited memory is sensitive to amnestic treatment. If MC memory and memory surviving cold-shock 2 h post training are equivalent, then delivering a cold-shock at least two hours after MC should not affect 24 h memory of the training. However, a cold shock delivered 2 h after MC was found to disrupt memory [25], suggesting that MC yields an additional memory type which consolidates slower than ARM since it is disrupted by cold shock. Therefore, this MC-elicited memory was termed Protein Synthesis-Independent Memory (PSIM) to differentiate it from bona fide ARM emerging after cold shock [25]. Whether PSIM represents a slow consolidating component of ARM elicited by the multiple training rounds, or a novel memory type is currently under investigation. Nevertheless, it appears prudent at this point not to assume that cold shock-resistant memory (ARM) and MC-elicited memories are identical, or that they require activity of common genes or engage common neuronal circuitry, unless they have been explicitly tested in both assays. It is possible that molecules affecting PSIM differentially will be identified in the future, including some known to play a role in cold-shock persistent ARM, but have not been cross-tested for MC-elicited memories. We have adopted the term MC-memory to refer to PSIM and the ARM-like memory elicited by massed conditioning to semantically differentiate it from memory elicited by one round of conditioning and resistant to cold shock, which we still refer to as ARM.

## 4. Distinct Properties of MC-Memory, ARM, and LTM

Even though MC-elicited memories, ARM, and LTM partially overlap in time, they differ in their time of emergence, duration, and functional characteristics (summarized in Table 1). As it is progressively formed around 30 min after training [16], ARM is detectable as early as 2 h after training, while LTM is still non-apparent. LTM becomes consolidated more gradually than ARM/MC memory being apparent no sooner than 6 h after training. At this point, MC-elicited memories overlap in time with LTM and may last for 24–48 h. In contrast, LTM is maintained for many days or even more than a week [5,6].

Although both LTM and MC-memories are elicited by repeated cycles of training, they present distinct characteristics in addition to the strict dependence of the former on protein synthesis. Typically, LTM lasts longer than 24 h, while MC-elicited memories appear to decay soon after 24 h. Its persistence, as well as its absolute dependence on new protein synthesis suggest that LTM is energetically costly. Indeed, absence of food after SC proves fatal shortly after completion of training. In contrast, viability is not affected significantly by lack of nourishment after MC [26,27].

Interestingly, although memory is known to decline with advanced age, LTM is affected differentially, while MC-memories appear intact [28,29]. Moreover, LTM and MC memories appear differentially affected by interactions within the group of trained and tested flies. SC-trained flies tested the next day remember the learned information either when tested individually or when tested in groups. However, flies after MC tend to remember better when tested in groups rather than on their own [30]. This indicates that MC recall relies on social interactions, or otherwise the group context in which training occurred. MC-elicited memories therefore may be more context-dependent than LTM. Interestingly, a novel context-dependent long-lasting memory described recently (cLTM) is also protein synthesis-independent like MC-elicited memories [31]. This suggests that context dependence at memory testing may generally characterize protein synthesis-independent memory forms. Such relationships may be elucidated if overlapping gene networks and molecular pathways are engaged for these two memory types in the same or distinct neurons.

**Table 1 ijms-23-08527-t001:** Main differences between ARM/PSIM and LTM.

	ARM/PSIM	LTM	References
training protocol	cold shock or massed training	spaced training	[6,17]
maintenance	<2 days	weeks	[6,17]
genes involved	e.g., *radish*	e.g., *CREB*	[17,18,32,33,34]
affected by inhibitors	pCPA, Fasudil	CXM	[6,35,36]
excess energy demand	no	yes	[26,27]
affected by aging	no	yes	[28,29]
affected by social context	yes	no	[30]

## 5. Molecular Mechanisms Differentially Engaged in ARM and MC-Elicited Memories

Do ARM and MC memories engage distinct molecular mechanisms than those requisite for LTM? In addition to the protein synthesis independence, evidence for engagement of distinct molecular mechanisms emerged from the abovementioned observation that mutations in the *radish* (*rad*) gene differentially affect ARM and MC memories, but not LTM [17,18,32]. The Radish (Rad) protein is present both in the cytoplasm and nucleus of CNS neurons [37] and based on sequence homology it is thought that it might act as a GTPase activator [38]. By this token, its most likely human orthologue is the GTPase-activating Rap/RanGAP domain-like 3 protein GARNL3. However, whether this putative Rad function contributes to ARM and, if so how remains unknown. In addition, Rad contains 23 potential target sites of Protein kinase A (Pka-C1) [32]. This is significant, because Pka-C1 mutants exhibit enhanced ARM (Table 2) probably due to impaired memory decay [39,40]. Thus, Rad may be a PKA substrate, but whether this in fact contributes to ARM and/or PSIM needs to be determined experimentally.

The Rad link to intracellular signaling was further supported by the observation that administration of p-chlorophenylalanine (pCPA), an inhibitor of serotonin synthesis, impairs ARM. Of the five serotonin receptors, 5HT1A is essential for ARM formation ([36] and Table 2). In confirmation, mutations in the *ddc* gene encoding the enzyme necessary for serotonin and dopamine biosynthesis also disrupt ARM ([36] and Table 2). Importantly, pCPA feeding to *rad* mutants does not result in additive ARM impairment, supporting the notion that serotonergic signals transduced via the 5HT1A receptor engage Rad in a signaling cascade essential for ARM [36]. Consistent with the results of *ddc* attenuation, impaired ARM emerged upon downregulation of the Dop2R dopamine receptor as well [41]. It remains unclear however whether Dop2R signaling requires Rad activity. In addition, attenuation of the pivotal enzyme for octopamine biosynthesis (a norepinephrine insect analog) TβH and one of the receptors it engages, Octβ2R, impairs ARM [42] in a Rad-independent manner.

It therefore appears that there are at least two parallel signaling pathways implicated in ARM, a serotoninergic one mediated via 5HT1A and Rad, and a Rad-independent octopaminergic cascade, possibly required in distinct neurons engaged in ARM formation, storage, or recall. Another possible player is likely Protein Kinase C (PKC), since a constitutively active form, PKM, enhances ARM independently of Rad [43]. This is consistent with either non-specific ARM enhancement or because it is downstream of Rad. Arguing for the non-specific case, PKC downregulation has been found to impair both LTM and ARM [34].

A novel molecular pathway engaged in ARM formation was suggested by the observation that mutants in the adaptor protein Downstream of Receptor Kinase (*drk*), or Growth factor Receptor Bound protein 2 (GRB2) in mammals, present defects both in cold-shock and MC-elicited memories in a Rad-independent manner [35]. Although Drk usually acts downstream of Receptor Tyrosine Kinases (RTKs) to activate the Ras/Raf/MAPK signaling pathway [44], the receptor engaged leading to ARM and MC-elicited memories remains unknown. Knockdown of the canonical effector of RTK activation Ras1 and Raf enhances ARM, indicating that the normal function of this signaling cascade is to suppress this memory form likely in favor of protein synthesis-dependent LTM [45].

Significantly, however, the ARM/MC memory deficit of *drk* mutants is reversible by activated Rho Kinase (Drok), suggesting signaling to actin polymerization/depolymerization. In confirmation of this hypothesis, filamentous actin was significantly reduced in *drk* mutants and the Drok inhibitor drug Fasudil impaired ARM and decreased filamentous actin levels in the CNS of control flies [35]. This predicts that additional members of the actin polymerization/depolymerization cascade will likely contribute to this pathway. If so, ARM may at least in part be mediated by the state of the actin cytoskeleton within relevant neurons of the fly CNS. Notably, the WASp actin nucleation factor and the Arp2/Arp3 complex required for polymerization of branched microfilament arrays are involved in ARM forgetting and their knockout enhances ARM [46], lending further credence to the actin polymerization as cardinal for ARM hypothesis.

**Table 2 ijms-23-08527-t002:** Phenotypic outcome of altered expression of genes involved in ARM.

Gene	Human Ortholog	Mutant/Reduced Expression/Downregulation ARM Phenotype	Overexpression/Upregulation ARM/MC Memory Phenotype	Flybase ID	Reference
*5-HT1A*	HTR1A	impairment		FBgn0004168	[36]
*brp*	ERC2	impairment		FBgn0259246	[20]
*CaMKII*	CAMK2D		impairment	FBgn0264607	[47]
*CASK*	CASK	impairment		FBgn0013759	[47]
*Cdc42*	CDC42	enhancement	impairment	FBgn0010341	[46,48]
*Ddc*	DDC	impairment		FBgn0000422	[36]
*dilp3*	INS	impairment		FBgn0044050	[49]
*dnc*	PDE4B	impairment		FBgn0000479	[50]
*Dop2R*	DRD2	impairment		FBgn0053517	[41]
*drk*	GRB2	impairment		FBgn0004638	[35]
*drok*	ROCK2			FBgn0026181	[35]
*Octβ2R*	HTR4	impairment		FBgn0038063	[42]
*Pka-C1*	PRKACA	enhancement		FBgn0000273	[39]
*Pkc98E*	PRKCE	impairment		FBgn0003093	[34]
*rad*	GARNL3	impairment		FBgn0265597	[17,18,32]
*Rgk1*	RRAD	impairment		FBgn0264753	[51]
*scrib*	LRRC1	enhancement		FBgn0263289	[52]
*Tβh*	DBH	impairment		FBgn0010329	[42]
*wasp*	WAS	enhancement	impairment	FBgn0024273	[46]
*arp2*	ACTR2	enhancement		FBgn0011742	[46]
*arp3*	ACTR3	enhancement		FBgn0262716	[46]
*mcu*	MCU	impairment		FBgn0042185	[53]
*micu1*	MICU1	impairment		FBgn0031893	[53]
*ras*	HRAS	enhancement		FBgn0003205	[45]
*raf*	BRAF	enhancement		FBgn0003079	[45]
*DopR*	DRD5	impairment		FBgn0011582	[54]

Additional genes that contribute to ARM/MC memory, with yet unclear roles in the abovementioned or additional molecular pathways are presented in Table 2. The presence of cAMP signaling members such as the phosphodiesterase Dnc and PKA-C1 is predicted by the engagement of serotoninergic and dopaminergic receptors in ARM. Another gene, *scribble* (*scrib*), whose disruption leads to enhanced ARM, is placed in a pathway downstream of the dopamine receptor Damb and acts through the Rac/Coffilin signaling pathway to mediate forgetting [52], again implicating actin polymerization in the process. However, the contribution of cytoskeletal dynamics and how it might regulate ARM/MC memories needs further confirmation and its mechanistic elucidation will likely require identification of additional proteins involved in the process.

Another significant issue that needs to be addressed systematically given the recent finding that MC yields both ARM and PSIM [25] is to determine whether extant mutants affect both processes or not. A number, but not all, of these mutants affect both processes (Table 3), and it is still unclear whether mutants or molecular pathways affect MC-elicited memories and which one, differentially. It follows therefore that mutants should be tested in both ARM-yielding protocols to ascertain their contribution and potentially identify PSIM-specific mutations within the extant pool or novel ones. Mutants specifically affecting PSIM are essential towards elucidation of potential molecular mechanisms that characterize it and differentiate it from ARM.

## 6. Neuronal Circuits Engaged in ARM/MC-Memory

What are the neurons in the adult Drosophila CNS essential for ARM? When known, the expression pattern of genes with validated contribution to ARM has led to RNAi-mediated adult-specific knockdown of the encoded proteins therein. Emergence of deficient ARM/MC memory verifies the contribution of these neurons, eliminates the possibility of a developmental origin for the deficit, and ascertains the role of the gene in the process(es). An alternative strategy involves the thermo-sensitive *shibire* (*shi^ts^*) transgene encoding Dynamin [55]. Acute exposure to 30 °C of flies expressing the *shi^ts^* transgene in specific neurons blocks neurotransmitter reuptake, essentially synaptically silencing them [56], enabling potential memory deficits to emerge. Both RNAi and synaptic silencing approaches are limited by the availability and expression specificity of “driver” strains. However, the already extensive arsenal of such drivers is being constantly expanded, making it unlikely not to find appropriate strains for such experiments.

The *rad*, *drk*, and *bruchpilot* genes are preferentially expressed in the Mushroom Bodies [32,56,57], where the CS and US information eventually converges [5]. The Mushroom Bodies (MBs) are bilateral clusters of neurons with their somata in the dorsal posterior of the adult brain, their dendrites extending ventrally to form the characteristic neuropils known as calyces and their axons projecting to the anterior of the brain. The MB neurons (MBNs) form three anatomically distinct groups, the α/β, α′/β′, and γ, which are also characterized by differential gene expression [58].

Initial studies on the *ala* mutant flies, which lack one or more MBN groups, revealed normal ARM if either α or β neurons were absent but not if both of them were missing [59,60]. Therefore the α/β neurons appear redundant for ARM. More recent studies have shown that the α/β and α′/β′ MBNs are crucial for ARM retrieval [61]. The main role for γ neurons in the process thus far is to receive dopaminergic input relating presentation of the US [54]. On the other hand, dopaminergic neurons of the Protocerebral Posterior Lateral (PPL1) cluster participate in ARM inhibition. Blocking neurotransmission from these dopaminergic neurons results in enhanced ARM. It appears likely then that these dopaminergic neurons mediate ARM inhibition to allow LTM formation [62]. The d5HT1A receptor on α/β neurons receives input from two Dorsal Paired Medial (DPM) neurons that secrete Serotonin, which is essential for ARM [36]. While the serotonergic DPMs innervate the α/β neurons, the octopaminergic Anterior Paired Lateral Neuron (APL) projects to α′/β′ MBNs and appears to regulate ARM independently of the DPMs [42].

Except for the MBs and their apparent input neurons, other neurons in the fly brain are also involved in ARM/MC memories. The gene *dunce* appears to be necessary not only in the MBs but also in antennal lobe (the insect olfactory bulb equivalent) local neurons (LNs) for normal ARM [50]. Furthermore, Insulin-producing cells (IPCs) seem to play a role in ARM by releasing dilp3, which binds to the Insulin Receptor (InR) of the Fat Body (FB) cells [49]. Apparently then, ARM/MC-elicited memories are governed by a complex neuronal regulatory machinery involving many parts of the fly brain. Whether all information eventually reaches the MBs, or some processes contribute to ARM in neurons outside the MBs processes remains to be elucidated.

## 7. Conservation of Drosophila ARM/MC Memory Characteristics and Molecular Components

Memory surviving amnestic treatment (ARM) was apparently first described in Drosophila and it is in this experimental organism that the molecular mechanisms and neuronal circuitry that supports it were defined. Although ARM per se has been verified in a couple more species, it is unreasonable to maintain that these elaborate molecular mechanisms and circuitry evolved to serve a particular type of memory specifically in Drosophila. So, are ARM or similar memories present in other species? The collective answer to this question is presented in Table 4.

Operationally, MC of the gill-withdrawal reflex response in the sea slug *Aplysia* resembles the effects of this type of conditioning in Drosophila, as it elicits short-lasting effects compared with the persistent consequences of SC [63]. A memory operationally closer to that surviving anesthesia in Drosophila has been described after aversive conditioning of the land slug, *Limax flavus.* A memory of the aversive association that is resistant to cold-shock emerges if the amnestic treatment is administered not immediately, but within 24 h post-training. Memory past that critical period appears fully consolidated and resistant to the insult [64]. Furthermore, similar to Drosophila, massed conditioning in appetitive or aversive olfactory conditioning in the nematode worm *Caenorhabditis elegans* yields transcription- and translation-independent memory in contrast to spaced training, which is disrupted by protein synthesis inhibition [65,66]. Interestingly, in *C. elegans*, MC-yielded memory is sensitive to cold shock administered soon after training in agreement with the abovementioned results in a similar experimental scenario in Drosophila [25].

Behavioral assays can be more elaborate in vertebrates than invertebrates because of size and better-understood behavioral repertoires. Nevertheless, as in Drosophila, mammals including humans are amenable to MC, which leads to shorter memory retention than memories elicited by SC. Moreover, in certain behavioral paradigms in mice, such as that of long-term adaptation of the horizontal optokinetic response (HOKR), MC-derived memory is clearly distinct from SC-derived memory in retention duration and dependence on protein synthesis [67]. Further experiments on synaptic plasticity induced by massed or spaced training revealed that SC leads to more acute changes in synapse formation compared to the massed protocol [68], as expected for protein synthesis-dependent memories. In support of such differential effects of massed versus spaced conditioning on synaptic strength, one ex vivo study reported that MC led to increased synaptic plasticity in Drosophila larval neuromuscular junctions (NMJs) [69]. These differential effects of MC in synaptic plasticity across species could be consequences of its link with changes in the actin cytoskeleton.

In fear conditioning, rodents are trained to associate an auditory cue with electric foot-shock. The effects of both massed and spaced training on fear memory are in accordance with the findings in Drosophila with respect to increased persistence of memories elicited via spaced than by MC and that SC is more effective for contextual fear conditioning compared with MC. In addition, administration of a protein synthesis inhibitor before fear conditioning in mice affects both the contextual and cued responses of the conditioned animals, irrespective of massed or spaced protocol. Significantly, there appears to be a translation-independent memory trace produced by MC, as shown by Long Term Potentiation (LTP) experiments in mouse hippocampal slices [70]. Furthermore, evidence from LTP studies in rats reveals differential effects of spaced stimulation in synapse recruitment [71]. It appears then that MC and spaced conditioning produce distinct memory types in vertebrates as well as in Drosophila. Perhaps additional evidence in support of ARM in vertebrates may be uncovered by careful analysis of protein synthesis-independent memory traces.

**Table 4 ijms-23-08527-t004:** Features that define ARM in Drosophila have been observed in other species. (ITI = Inter-trial Interval).

Class	Species	Cold-Shock Resistance	Effect of ITI on Behavior	Protein-Synthesis Independence Following MC	Assay	References
Invertebrates						
*Sea Slug*	*Aplysia*	-	longer-lasting habituation by SC	-	habituation (gill-withdrawal reflex)	[63]
*Slug*	*Limax Flavus*	memory resistant to cooling	-	-	aversive conditioning	[64]
*Crab*	*Chasmagnathus*	-	longer-lasting habituation by SC	-	protein synthesis-independent memory trace	[72,73]
*Worm*	*Caenorhabditis elegans*	memory resistant to cooling	longer-lasting memory by SC	protein synthesis-independent memory	appetitive or aversive conditioning	[65,66]
*Bee*	*Apis mellifera*	-	longer-lasting memory by SC	transcription-independent short-term memory	appetitive conditioning	[74]
Vertebrates						
*Pigeon*	*Columba livia*	*-*	more robust response and better retention by SC	*-*	appetitive conditioning	[75]
*Mouse*	*Mus musculus*	-	longer-lasting memory by SC	protein synthesis-independent memory	long-term HOKR adaptation	[67]
		-	-	no translation-independent memory	fear conditioning	[70]
*Rat*	*Rattus norvegicus*	-	longer-lasting memory by SC	-	radial maze learning	[76]
*Human*	*Homo sapiens*	-	longer-lasting memory by SC	-	various cognitive tasks	[77,78,79]

Human studies also suggest differential effects of spaced and massed conditioning on memory, in support of the positive impact of spaced intervals between training sessions on memory retention (reviewed in ref. [77]). While massed training leads to better performance immediately after learning, spaced training favors performance at later times [61]. It has been estimated that this distinctive outcome involves mechanisms that are present during early training known as working memory [80], defined as fore-front maintenance of information for a brief amount of time [81]. Interestingly, evidence suggests a correlation of massed, but not spaced, conditioning with working memory capacity [78,79]. Whether ARM by analogy represents a manifestation of working memory in Drosophila remains to be experimentally explored.

An important correlation regarding the nature of ARM emerged from studies of molecular mechanisms engaged in amygdala-dependent fear conditioning in rats. These studies revealed a significant common characteristic between ARM in Drosophila and fear memories at the molecular level. Cue-dependent fear conditioning leads to GRB2-mediated signaling to p190 RhoGAP-downstream kinase (ROCK) in Lateral Amygdala (LA) [82]. These are the rat orthologs of DRK and Rho kinase (Drok) in Drosophila, which are established components of an ARM/MC-memory-mediating molecular pathway(s) in the MBs [35]. Therefore, it appears that fear conditioning mediating processes in rodents, at least from a molecular perspective, are likely closely related to those that govern ARM in Drosophila.

An additional potential link with fear conditioning in rodents and ARM in Drosophila was provided by a rather impactful recent study. Aversive SC in Drosophila was shown to actually yield two concurrent complementary memories, an avoidance memory of the punished CS+ and a “safety memory” of the unpunished CS-. In fact, the avoidance memory was insensitive to protein synthesis inhibitors, did not depend on spaced conditioning trials, and decayed faster than “safety memory”. Therefore, there are multiple aspects of avoidance memory that resemble ARM and in fact may be linked to fear of the predicted punishment. Whether ARM is at least in part a manifestation related to fear conditioning in vertebrates remains to be determined. Shared components of the molecular cascades involved in Drosophila ARM/MC memories and fear conditioning in rodents are likely to provide initial such correlations. Whether Drosophila mutants in *rad, drk, drok,* or *rhoGAP* present impaired avoidance memory but normal “safety memory” remains to be experimentally determined. If so, this will provide much-needed insight into the nature of ARM and its relation to vertebrate memories.

## 8. Perspectives

Despite significant advances in elucidating molecular components and neuronal circuitry engaged in ARM, it is apparent that it is still only defined in Drosophila, though it is highly unlikely to operate in this insect alone. Its characteristic translation-independent low energetic cost [26,27] and speedy consolidation that render it resilient to insults such as cold shock strongly support its utility. If ARM is indeed differentially linked to memory of punishing stimuli, then its value for survival is rather obvious. Its emerging links with cytoskeletal and synaptic plasticity provide a novel mechanism of expedient memory consolidation with significant advantages for survival, which will enhance understanding the nature of ARM and its vertebrate counterpart(s).

Establishment of the anticipated link or correlation to vertebrate memories will broaden the scope of ARM/MC memories research. Drosophila, with its extensive molecular and genetic arsenal and continuously enriched behavioral repertoire, is still uniquely poised to reveal novel genes, signaling pathways, cellular mechanisms, and neuronal circuits underlying ARM. Such information will likely prove invaluable towards understanding this novel memory type in vertebrates and reveal potential links to human pathologies affecting cognition.

## Figures and Tables

**Figure 1 ijms-23-08527-f001:**
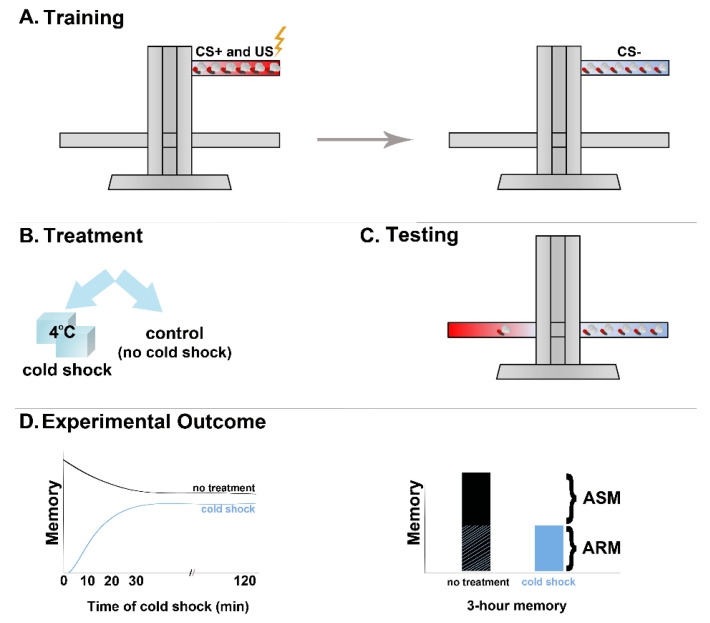
Aversive olfactory association paradigm in Drosophila. (**A**) Training of flies consists of pairing an odor (CS+) with electric foot shock (US) followed by presentation of another odor (CS-) in the absence of shock. (**B**) Treatments, such as cold shock, may be employed after training. (**C**) Testing of the performance for the previously learned association involves simultaneous presentation of the CS+ and CS- odors for the flies to choose. (**D**) Calculation of the performance index results in a representation of memory retention. In the case of cold shock treatment two hours post training, MTM loses its ASM counterpart and consists only of ARM.

**Figure 2 ijms-23-08527-f002:**
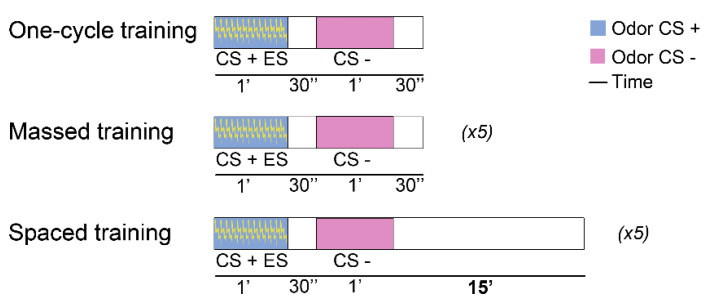
Standard training protocols for aversive olfactory association conditioning in Drosophila. The difference between the two multiple-cycle training protocols, massed and spaced, lies in the absence of a fifteen-minute resting interval.

**Table 3 ijms-23-08527-t003:** Genes involved in ARM have been tested in either or both protocols that yield ARM, and their role has been identified in specific neuronal subsets.

Gene	Cold Shock Protocol	Massed Protocol	Neurons/Cells	Reference
*5-ht1a*	x		α β KCs, DPM	[36]
*brp*	x		MBs	[20]
*camkII*		x	α′ β′ KCs	[47]
*cask*		x	α′ β′ KCs	[47]
*ddc*	x		DPM	[36]
*dnc*	x		LNs, (MBs)	[50]
*dop2r*	x		DANs, LNs, APLs, αβ and γ KCs	[41]
*drk*	x	x	αβ KCs	[35]
*drok*	x	x	αβ KCs	[35]
*dilp3*		x	IPCs	[49]
*mcu*	x	x	MBs	[53]
*micu1*	x	x	MBs	[53]
*pka-c1*	x	x	MBs	[39]
*rad*	x	x	MBs	[17,18,32]
*rgk1*	x		MBs	[51]
*scribbled*	x			[52]
*pkc*		x		[34]
*cdc42*	x	x		[46,48]
*wasp*	x	x		[46]
*arp2*	x			[46]
*arp3*	x			[46]
*ras*	x	x	γ KCs	[45]
*raf*	x		γ KCs	[45]
*Octβ2R*	x		α′ β′ KCs	[42]
*Tβh*	x	x	APLs	[42]
*DopR*		x	γ KCs	[54]

## Data Availability

Not applicable.

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
