# Peer review of "Anesthesia Resistant Memories in Drosophila, a Working Perspective"

_ijms, 2022, doi:10.3390/ijms23158527_

Round 1
Reviewer 1 Report
Bourouliti and Skoulakis present here a very thorough, well-written and enjoyable review on the fascinating, still somehow under-researched anaesthesia resistant memory and protein-synthesis independent memory in Drosophila (ARM and PSIM). I want to acknowledge the effort that the authors put in recapitulating the state of the art in the field, the smart use of tables and the highlight of open questions remaining in our understating of protein-synthesis independent memories.
I remember wondering about the definite prove that cold-shock resistant memory and Mass-training induced memory where, as assumed in the field at the time, the same physiological phenomenon. Hence, I find the topic of this article extremely interesting, and I will be very happy to see it published.
I have a few suggestions that I hope increase the clarity of some of the main concepts in his paper, and some highlights of what I consider to be typos:
- line 34: I would not consider 90V a mild shock for a fly, maybe the authors could rephrase that
- Figure 1D- there seem to be issues with the download version of the pdf, the graph looks empty
- Line 94: Knapek et al (2011) J Neurosci 31:3453-8 (ref 41 in the paper) should also be cited here as the first report (to my knowledge) of immediate parallel formation of ARM
- Line 210- PKM typo
- Line 279: the sentence “Therefore the α and β neurons appear redundant for ARM” is misleading as it can be understood as α and β being different neurons, I would rephrase it as “the α and β lobes”
- Finally, I appreciate the comparative approach of the last section, but might consider rephrasing the title or include a disclaimer on the different connotations of ARM. By definition, ARM speaks of anaesthesia resistant memory, and although one of its characteristics is to be protein-synthesis independent, those concepts, as well-stablished throughout the manuscript, are not interchangeable. Most of the instances in this section in vertebrates are indeed protein-synthesis independent forms of memory, or massed vs spaced training-evoked, but not necessarily anaesthesia-resistant.
Author Response
Reviewer 1.
line 34: I would not consider 90V a mild shock for a fly, maybe the authors could rephrase that
this was rephrased, though 90V DC is not such a strong electric shock even for us humans. One can hardly feel it
- Figure 1D- there seem to be issues with the download version of the pdf, the graph looks empty
Thanks for pointing this out. Not sure what happened. Hopefully it is fixed in the new version
- Line 94: Knapek et al (2011) J Neurosci 31:3453-8 (ref 41 in the paper) should also be cited here as the first report (to my knowledge) of immediate parallel formation of ARM
done
- Line 210- PKM typo
PKM is the constitutively activated cleaved formed of aPKC I believe. If not so we will be happy to change it.
- Line 279: the sentence “Therefore the α and β neurons appear redundant for ARM” is misleading as it can be understood as α and β being different neurons, I would rephrase it as “the α and β lobes”
Rephrased as α/β neurons
- Finally, I appreciate the comparative approach of the last section, but might consider rephrasing the title or include a disclaimer on the different connotations of ARM. By definition, ARM speaks of anaesthesia resistant memory, and although one of its characteristics is to be protein-synthesis independent, those concepts, as well-stablished throughout the manuscript, are not interchangeable. Most of the instances in this section in vertebrates are indeed protein-synthesis independent forms of memory, or massed vs spaced training-evoked, but not necessarily anaesthesia-resistant.
Section title changed, hopefully it is more appropriate now
Reviewer 2 Report
In their review, “Anesthesia Resistant Memories in Drosophila, a working perspective”, Bourouliti and Skoulakis present a clear and organized discussion on some of the intricacies of Anesthesia Resistant Memory (ARM) in Drosophila. There are only a few places where additional attention seems necessary. I have organized my comments below in order that they appear in the review.
Line 51: “or else” is uncommon phrasing in English. I recommend, “also known as”
Line 58: “probably soon after the learning episode.” This dangles as conjecture and is not supported by references. Although this point is taken up later, some support for this conjecture here seems to be required.
In the paragraph starting on line 62: The authors bring up the important point that ARM is isolated by cold shock and this cold shock may not be a bone fide anesthetic. Since anesthesia is central to the review topic, and cold shock is the only means for generating ARM stated, the authors need to explain further and define these central concepts. Specifically, please address the following questions:
11) What defines an anesthetic?
22) What is known and unknown about the role of the cold treatment and how it relates to anesthesia?
33) What other anesthetics have been used successfully to isolate ARM in Drosophila?
Figure 1B. Treatment – this graphic is confusing. In one path there are what looks to be ice cubes and 0C, but the other is blank – the controls flies must be treated in some way that can be indicated by a graphic - or at least the word control can be written. Also, is the cold treatment 0 ⁰C or 4 ⁰C as stated in the text (line 63)?
Figure 1D lines are bars indicating performance levels are missing from the graph. In the 3-hour bar chart, could the ARM and ASM be separately indicated by hatching rather than brackets or in addition to the brackets– to make clear the additive memory processes present in control treatment animals?
Table 1: The absence of energy cost in ARM is speculative and could be better phrased. I also believe another column containing supporting references would be very helpful for the reader.
Line 349: working memory comparison. If there is a serious comparison to be made between ARM and working memory, that is not simply because working memory does not last as long as the necessary Intertrial intervals, then this should be explored more. At the very least, working memory should be defined, and the different effects of massed and spaced training on working memory need to be stated.
Line 365: “ARM n Drosophila” should be “in”.
Line 508: the Wimmer reference now exists as a J. Neuro paper, which should be used in this review.
Author Response
Reviewer 2.
Line 51: “or else” is uncommon phrasing in English. I recommend, “also known as”
Fixed, thank you
Line 58: “probably soon after the learning episode.” This dangles as conjecture and is not supported by references. Although this point is taken up later, some support for this conjecture here seems to be required.
That is why we use the word probably. The exact timing may be unknown, but it does happen after learning I would venture. I would be happy to discuss further clarifications if needed
In the paragraph starting on line 62: The authors bring up the important point that ARM is isolated by cold shock and this cold shock may not be a bone fide anesthetic. Since anesthesia is central to the review topic, and cold shock is the only means for generating ARM stated, the authors need to explain further and define these central concepts. Specifically, please address the following questions:
11) What defines an anesthetic?
Thanks for pointing this out. We have added a (general) definition
22) What is known and unknown about the role of the cold treatment and how it relates to anesthesia?
We hope that this point was also addressed satisfactorily
33) What other anesthetics have been used successfully to isolate ARM in Drosophila?
Sadly, none we are afraid and we state this
Figure 1B. Treatment – this graphic is confusing. In one path there are what looks to be ice cubes and 0C, but the other is blank – the controls flies must be treated in some way that can be indicated by a graphic - or at least the word control can be written. Also, is the cold treatment 0 ⁰C or 4 ⁰C as stated in the text (line 63)?
We changed the temperature to 4C, thank you and added text for control
Figure 1D lines are bars indicating performance levels are missing from the graph. In the 3-hour bar chart, could the ARM and ASM be separately indicated by hatching rather than brackets or in addition to the brackets– to make clear the additive memory processes present in control treatment animals?
See above. Not sure why the conversion eliminated that line.
Table 1: The absence of energy cost in ARM is speculative and could be better phrased. I also believe another column containing supporting references would be very helpful for the reader.
Done as suggested
Line 349: working memory comparison. If there is a serious comparison to be made between ARM and working memory, that is not simply because working memory does not last as long as the necessary Intertrial intervals, then this should be explored more. At the very least, working memory should be defined, and the different effects of massed and spaced training on working memory need to be stated.
We have added text to that effect. Granted, that this is speculation based largely on phenomenology, but the evidence supporting this speculation is largely lacking and the statement is meant more as “food for thought”
Line 365: “ARM n Drosophila” should be “in”.
Fixed.
Line 508: the Wimmer reference now exists as a J. Neuro paper, which should be used in this review.
We could not find a J Neuro paper with the same authors and title. However we did find one with these specifications in “Memory and Cognition” (reference 79).
Wimmer, G.E. and R.A. Poldrack, Reward learning and working memory: Effects of massed versus spaced training and post-learning delay period.Memory & Cognition, 2022. 50(2): p. 312-324